# Mapping Area Changes of Glacial Lakes Using Stacks of Optical Satellite Images

Varvara Bazilova [1,2,*] and Andreas Kääb [1]

1 Department of Geosciences, University of Oslo, P.O. Box 1047, 0316 Oslo, Norway
2 Department of Physical Geography, Utrecht University, P.O. Box 80115, 3508 TC Utrecht, The Netherlands
* Correspondence: v.bazilova@uu.nl

**Abstract:** Glacial lakes are an important and dynamic component of terrestrial meltwater storage, responding to climate change and glacier retreat. Although there is evidence of rapid worldwide growth of glacial lakes, changes in frequency and magnitude of glacier lake outbursts under climatic changes are not yet understood. This study proposes and discusses a method framework for regional-scale mapping of glacial lakes and area change detection using large time-series of optical satellite images and the cloud processing tool Google Earth Engine in a semi-automatic way. The methods are presented for two temporal scales, from the 2-week Landsat revisit period to annual resolution. The proposed methods show how constructing an annual composite of pixel values such as minimum or maximum values can help to overcome typical problems associated with water mapping from optical satellite data such as clouds, or terrain and cloud shadows. For annual-resolution glacial lake mapping, our method set only involves two different band ratios based on multispectral satellite images. The study demonstrates how the proposed method framework can be applied to detect rapid lake area changes and to produce a complete regional-scale glacial lake inventory, using the Greater Caucasus as example.

**Keywords:** glacial lakes; glacial lake drainages; water body mapping; earth engine; multispectral images

## 1. Introduction

The cryosphere, and glaciers in particular, are very vulnerable to the current climatic changes and are important climate change indicators [1–3]. Glacial lakes are bodies of water that are located in glacier environments where the drainage is prevented by ice, bedrock or sedimentary moraine barriers [4]. Glacier lakes can be found under glaciers (subglacial), on glaciers (supraglacial), within glaciers (englacial) or marginal to glaciers (proglacial). In response to glacier retreat since the Little Ice Age (LIA) and in especially in recent decades, different types of ice-marginal and moraine-dammed lakes have been growing (e.g., [4–6]).

The hazards that are associated with glacial lakes are often part of a cascade of events, for example triggered by ice- or rock-avalanche impacts and/or leading to devastating debris flows (e.g., [7,8]). Glacier lake outburst floods (GLOFs) are one of the most common hazards associated with glacial lakes. GLOF refers to a sudden release of water and sediments from a lake (glacier-dammed, moraine-dammed, or any other type). These floods heavily affect close-by communities, but are also the most far-reaching glacier-related hazard, sometimes affecting land hundred kilometers and more downstream of the failed glacial lake. Among other hazards, GLOFs result in the release of large amounts of sediments that travel downstream and can damage, for instance, hydropower and aquaculture infrastructure and agricultural land (e.g., [3,9–12]).

General overviews, based on GLOFs reported in the literature and regional inventories since the 19th century concluded that GLOFs seem not to be directly influenced by the changing climate and the response of glacial lake behavior might be delayed in time with respect to glacier changes [13]. In contrast to mentioned global-scale studies, [14] proposed that in the Himalaya the average annual frequency of outburst floods has no credible trend.

Due to the lack of on-site information, the knowledge about GLOFs that have occurred in remote areas is limited. The global assessments that have been performed are based on the societal and economical impact that GLOFs had caused [15]. Therefore, any inventory of GLOFs that is based on on-site reports is inherently incomplete and uncertain. The temporal distribution of the GLOFs reported in the literature is not consistent either and changes due to inconsistency of the research practices during different epochs and the growing research interest in glaciers [11]. Although a number of studies reported global [6] and regional (e.g., [16–18]) glacial lake expansion, there is only limited evidence for changes in frequency of GLOFs. By compiling a global GLOF database [11] also challenges the opinion that more GLOFs occur under the warming climate. Still, GLOF response to climate change remains poorly understood [7]. Besides the unknown pattern of global response of glacial lake hazards, the water stored in the glacial lakes is often neglected in models of the hydrological response to glacier mass loss [19].

Currently, glacial lakes hold about 0.43 mm of sea-level equivalent globally [6]. However, there are uncertainties in these estimations. For example, up to 56% of the lakes at the margin of the Greenland Ice Sheet could be unaccounted for global volume and change estimates [20]. As the number and size of glacier lakes have increased dramatically in individual regions and worldwide [6], accurate re-mapping of glacial lakes at regular time intervals appears necessary.

For mapping glacial lakes, water surface detection using multispectral satellite images is most often used, but frequently suffers from problems specific to high-mountain areas such as terrain and cloud shadows, water turbidity, or snow and ice cover. In this study we refine methods that are commonly used for water mapping, specifically addressing problems that are associated with high mountain and glacier environments. We first present the data we used for this study and mention existing methods. Then we suggest a set of methods that can be used for different applications related to mapping glacier lakes and their short-term and long-term changes and demonstrate their performance. We consider multiple scenarios, where these methods can be applicable: short-term area changes as an initial indicator of a possible lake outburst (in case of substantial area loss), or dammed lake (in case of substantial area gain), respectively, and compiling a regional-scale inventory of glacial lakes to assess long-term changes. The purpose of this study is not to come up with one fixed workflow or a dataset, but rather to discuss a framework of methods that can be combined for different application scenarios. We address the question; which strategies can be applied for repeat glacial lake mapping based on stacks of openly available medium-resolution multispectral satellite data and on different scales? We use Landsat satellite data for this purpose as this program provides the longest satellite time series for assessing the long-term changes of glacial lakes.

## 2. Data

In addition to ground-based remote sensing methods (such as, for example, multibeam echosounders or bathymetric lidars) or airborne remote sensing methods (such as airborne lidar), satellite remote sensing is widely used for large-scale mapping of landscape features, including glacial lakes. Different types of satellite data can be used for glacial lake mapping. While multispectral optical images, as have been used in this work, represent the most common data type that is used for water mapping, there are other methods that do not rely on optical remote sensing: for example Digital Elevation Models (DEM) sink detection (e.g., [21]), Synthetic Aperture Radar (SAR) data (e.g., [22]) or such methods combined with optical data (e.g., [20]). DEMs are also often used to complement optical remote sensing data (e.g., [6,17,20,23]).

### 2.1. Optical Images

For the purpose of the present study, high-resolution multispectral optical satellite data from the Landsat 5 (Thematic Mapper—TM), Landsat 7 (Enhanced Thematic Mapper— ETM) and Landsat 8 (Operational Land Imager—OLI) satellites has been used. Consistent

near-global time-series of multispectral Landsat images are available since the launch of Landsat 5 in 1984 and have a spatial resolution of 30 meters with a 16 day revisit period, making the data well suitable for long-term monitoring of glacial lakes. Optical satellite images are recorded from space in the visible, Near Infrared (NIR) and Short-wave Infrared (SWIR) parts of the electromagnetic spectrum, ranging from 0.45 μm to 2.3 μm. Landsat mission sensors also include thermal infrared (TIR) bands (10.40–12.50 μm) that record long-wave surface emission. Landsat 5 and 7 have a lower radiometric resolution than Landsat 8 (8 bit and 12 bit, respectively). The 8-bit resolution frequently results in saturated visual bands over bright snow and cloud-covered pixels, a problem that is typically overcome in Landsat 8 12-bit data.

While it is clear that using Sentinel-2 data for the purpose of our study would improve the spatial resolution and thus mapping of particular small lakes, this advantage would come at the cost of a much shorter time series (30 years shorter than Landsat) that inhibits the mapping of long-term lake area changes. For short-term lake area changes such as lake drainage the higher temporal resolution of the Sentinel-2 constellation can certainly be an advantage for better estimating the exact date of a drainage, but we argue that the area changes associated with lake drainage or damming last typically long enough to be also detectable using Landsat revisit time. Regarding short-term lake area changes it is our aim to find indication of a drainage or damming, not to estimate the event data as accurate as possible. Combination of Landsat and Sentinel-2 data in one workflow was beyond the focus of the present study but we aim to assess a general methodology that should be adaptable to other Landsat-type data, such as from Sentinel-2.

In this study the entire archive of Landsat 5 imagery was used (availability depends of the area of interest and lasts until the end of the mission in 2012); Landsat 7 was considered suitable for glacial lake mapping when acquired between 1999 and 2002 (due to Scan Line Corrector—SLC—failure after May 2003 [24]); Landsat 8 data used in our study range from 2013 until the time of writing (2021). We used Landsat surface reflectance data, meaning that atmospheric scattering was corrected for in the top-of-atmosphere reflectances recorded by the satellite sensors. For Landsat 5 and Landsat 7 atmospheric correction, the LEDAPS algorithm [25] is used. Landsat 8 surface reflected images are produced using LaSRC [26]. Surface reflectances represent values that can be compared across different regions, which is important especially for high-latitude and high-altitude environments where the atmosphere and topographic variations such as slope magnitude and azimuth have a significant effect on solar irradiance [27]. All Landsat data are preprocessed and available as a Google Earth Engine (GEE) [28] image collection in its data catalog.

*2.2. Digital Elevation Models*

Along with optical Landsat images, we used digital elevation models (DEMs) to evaluate mapping methods. The DEMs used are the SRTM DEM (Shuttle Radar Topography Mission Digital Elevation Model) and TanDEM-X DEM. Both products are based on synthetic aperture radar (SAR) data (C-band and X-band). The SRTM DEM is a near global digital elevation model [29]. The SRTM was flown on board NASA space shuttle Endeavour in 2000. The horizontal resolution of the data used here is 3 arcseconds, which corresponds to about 90 meters. In this study the hole-filled SRTM for the globe Version 4 [30] was used as downloaded from the GEE catalog. TanDEM-X DEM is a near-global DEM, developed by German Aerospace Center (DLR) and Airbus Defence and Space under the TanDEM-X mission. The DEM is built from the data that have been acquired between 2010 and 2014 [31]. The mission provides multiple products. In this study we used the freely available TanDEM-X 90m version, meaning that the horizontal resolution of the used DEM is 3 arcseconds, which corresponds to 90m at the equator [32].

### 3. Methodology

#### 3.1. General Concepts

Multispectral optical remote sensing images are widely used for surface water monitoring (e.g., [33–35]) as well as monitoring of glaciers, glacier extent and snow cover (e.g., [27,36,37]). Multiple methods of automatic extraction of glacier and water outlines from multispectral images are widely described in the above literature. Most of the existing methods of automatic water and glacier detection utilize the spectral properties of the water, in particular the difference in its reflectance between the visible and infrared parts of the spectrum.

Various methods use multiple visible and infrared bands, such as normalized difference indices or ratios between bands. Among the most commonly used methods are the indices that are aiming either to highlight the water by enhancing the contrast of the image, or to exclude the water by highlighting other information (e.g., vegetation). All normalized difference indices use two wavelength bands where the surface of interest has a high difference in reflectance. "Normalization", dividing the difference in reflectance by its sum, allows to reduce the noise and account to some extent for different solar zenith angles, atmospheric conditions, and illumination conditions. The most commonly used normalized difference indices are normalized difference vegetation index (NDVI) [38], normalized difference water index (NDWI) [39], and normalized difference snow index (NDSI) [40,41], which is also sometimes called modified normalized difference water index (mNDWI) [42].

Multispectral vegetation mapping is based on the chlorophyll pigment properties, which strongly absorb visible light for use in photosynthesis. The ratio or the normalized difference allows to evaluate how green an area is, using the red and NIR bands (Equation (1)). Unless abundant phyto-plankton is present, which is rare in glacial lakes, the NDVI values for glacial water are typically low, because of the absence of the chlorophyll pigment in water. This usually helps to exclude false classification of dark areas that are covered with vegetation, and which are not water. However, in many cases the NDVI alone is not sufficient for water mapping because the low values not only can correspond to water, but also to rocks, or areas covered with snow or ice.

$$NDVI = \frac{(Xnir - Xred)}{(Xnir + Xred)} \tag{1}$$

where *Xnir* and *Xred* correspond to reflectance of NIR and Red bands of the multispectral optical images.

For water mapping, the normalized difference water index (NDWI) (Equation (2)) is often used. It takes advantage of the difference in water reflectance and absorption in green and NIR parts of the spectrum [39]:

$$NDWI = \frac{(Xgreen - Xnir)}{(Xgreen + Xnir)} \tag{2}$$

where *Xgreen* and *Xnir* represent reflectance of Green and NIR bands of the multispectral optical images. It is important not to confuse this index with another index with the similar name that has been proposed for remote sensing of liquid water and uses the NIR and SWIR bands for that purpose [43].

The modified normalized difference water index (mNDWI) Equation (3) is very similar to the NDWI, but uses SWIR instead of NIR [42]:

$$NDSI = \frac{(Xgreen - Xswir)}{(Xgreen + Xswir)} \tag{3}$$

where *Xgreen* and *Xswir* are reflectance values of Green and SWIR bands of the multispectral optical images.

The same index is often referred to as normalized difference snow index (NDSI) [40,41], which is used for snow mapping because of the enhanced contrast of water, ice and snow. This makes it difficult to use the mNDWI for water mapping in mountain areas where glacier- and snow-covered parts of the landscape are present at the same time.

Similar to the normalized indices, the ratio between bands can be used without normalization. The band ratios are commonly used for mapping glaciers and have been preferred over normalized difference indices for this purpose (e.g., [36,44,45]). They allow for finer class details to be visible in the result image and highlight water features better by enhancing the contrast between different values (Figure 1) Such band ratios were thus used in this study in addition to normalized difference indices (similarly to [17]): Ratio image of bands, used for the NDVI

$$R_{vegetation} = \frac{Xnir}{Xred} \tag{4}$$

Ratio image of bands, used for the NDWI

$$R_{water} = \frac{Xgreen}{Xnir} \tag{5}$$

Ratio image of bands, used for the NDSI

$$R_{snow} = \frac{Xgreen}{Xswir} \tag{6}$$

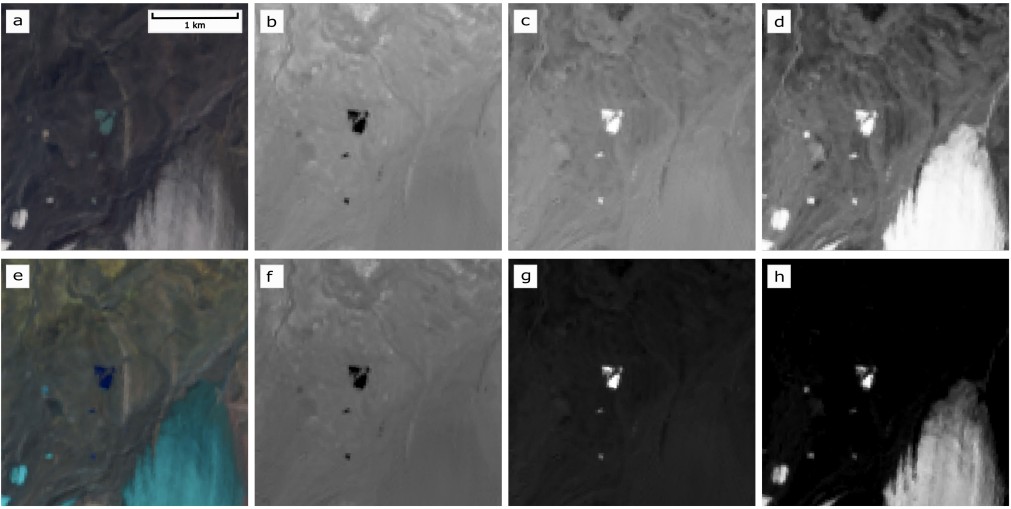

**Figure 1.** Examples of spectral indices and ratios (Landsat 8 image from 26/08/2020; Mount Elbrus, Central Caucasus). (**a**) Natural Color Composite (Red-Green-Blue), (**b**) NDVI, (**c**) NDWI, (**d**) NDSI, (**e**) False Color Composite (SWIR-NIR-Red), (**f**) NIR/Red ratio, (**g**) Green/NIR ratio, (**h**) Green/SWIR ratio.

### 3.2. Detection of Short-Term Changes in Image Time-Series

Due to the nature of the phenomena, monitoring lake drainage or damming requires sub-seasonal resolution of the data that are used for detection. Analysis of a time series of individual images allows to capture the intra-seasonal changes of the lake water level and area, and the sudden lake area change associated for instance with a lake drainage (Figure 2). The temporal resolution of the Landsat image acquisition is 16 days, therefore, under cloud-free conditions, it allows to identify such events within this timeframe.

In this branch of our method framework, the water pixels on each image are classified individually using the normalized difference water index (NDWI) and the blue band of the multispectral image with a manually defined threshold, similar to e.g., [6,17].

$$Water = NDWI \geq 0.24 \,\&\, (700 \leq Xblue \leq 2500) \tag{7}$$

The such-classified water pixels are added to original images as a new band, containing binary values (1—water, 0—no water). Similarly to [33] "water occurrence" information, these bands are aggregated together as a sum, meaning that the bigger the value is the more times a pixel is classified as water. The next steps can be divided into 2 major parts: (i) to define, if a drainage happened, (ii) to identify the time-frame of the event (Figure 2).

To examine the changes in the glacial lake area, the pattern of the aggregated water-pixel layer can be used. The changes in the lake area are indicated in the frequency of water observed at the particular pixel location. Therefore, if the lake has drained over the aggregated time period, pixels of the drained part of the lake will be classified as being less frequent water compared to the remaining lake part. The same applies for lake growth (and lake level increase), for instance in case of a landslide or surge damming, where newly inundated lake parts will have fewer water pixels aggregated over time than the pre-existing lake sections, if a lake existed at the location at all.

To find out in which time period a lake burst out, the time series of area changes can be examined. In the Google Earth Engine environment, the area time series curve is built across all the available images that are used for the analysis. This is done by counting pixels inside the polygon around the lake, that is defined manually. We assume that the sudden lake level change that is associated with a drainage results in a sudden drop in lake area.

### 3.3. Annual Image Stack Composites for Lake Mapping

The use of time series was proposed for glacier mapping by Winsvold et al. [37]. This includes calculating statistical values for each pixel (corresponding to the same location) of the georeferenced time series of images stacked together. In contrast to the above method (Section 3.2), the sequence of images plays no role in the method used in the following method. Extracting the minimum values of an image stack (e.g., minimum reflectance) allows to exclude seasonal snow cover or icebergs originated from calving [46]. A similar approach can be used for lake mapping using spectral indices instead of original bands (Figure 2). One synthetic mapping scene per year is created as composite from all Landsat images available for the area of interest. First, we compute indices and ratios for every scene, and then create a composite image. This image consists of statistical values calculated from the original Landsat image bands as well as calculated statistical values for the computed above 3 indices and 3 ratios.

The use of the minimum and maximum stack values for creating the mapping scene out of the image stack or index stack is not a robust way in itself and requires some additional processing to cope with image errors. Due to the Landsat 5 and Landsat 7 radiometric resolution (8 bit) and known Landsat technical issues (e.g., Transient Detector Failure or Sun Glint Anomaly) there are some errors in the Landsat satellite scenes affecting the composite values. The minimum value might be an outlier, generated by data issues, therefore filtering is needed.

For the purpose of this study the minimum values of the stack were removed before calculating the statistics (meaning that the remaining minimum value is the original 2-nd minimum value of the stack). The other source of noise and errors associated with the radiometric resolution of the sensors are saturated pixels of bright surfaces, such as snow and ice. As a general rule, the maximum values of the image bands are corresponding to the pixels covered by clouds, however in some of the areas Landsat 5 (TM) and Landsat 7 (ETM) scenes have high values, corresponding to the areas covered with snow. These pixels are excluded by masking pixels where the Blue band value ($X_{blue}$) equals 20,000.

The final data cube consists then of a series of annual composite images that each include the minimum and maximum values of the image bands and computed spectral indices and band ratios (NDVI, NDWI, NDSI, NIR/Red, Green/NIR, Green/SWIR). Here, this data cube was used to produce the glacial lake inventory for the Greater Caucasus mountain range.

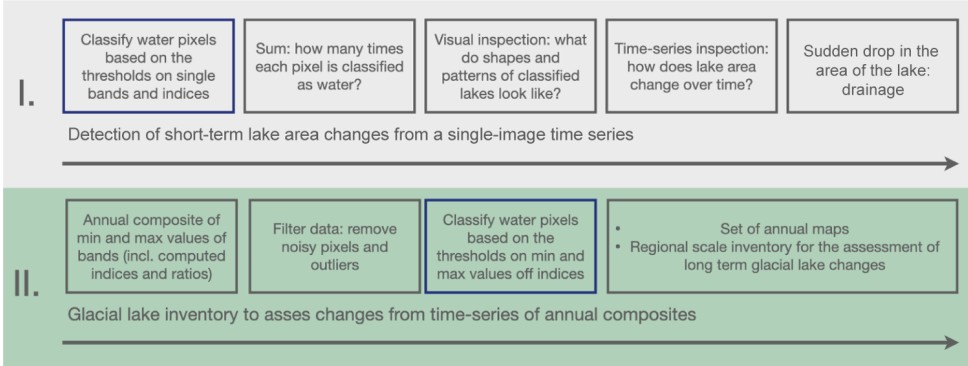

**Figure 2.** The core of the workflow for the proposed methods. (**I.**) identifying fast lake changes in a series of images; (**II.**) Compiling glacial lake inventory for glacial lake change assessment. Blue boxes indicate the "water mapping" step

### 3.4. Implementation: Glacial Lake Inventory

This section describes the implementation of the method of annual composite data cubes (Section 3.3) to compile a regional-scale inventory of glacial lakes. The glacier lakes were mapped on every time step (i.e., annual). These annual water maps were then merged into 5-year average layers to keep our results consistent with the existing global glacial lake inventory by [6]. Specifically, the annual maps were merged using the `.mean()` GEE command into 5-year average raster layers in the GEE-environment and then exported to ArcMap for postprocessing and further analysis. The postprocessing involved the conversion of the GEE raster results to vectors, calculating the area of the polygons and noise filtering.

To filter the noise that comes from misclassification of the images we used an area size threshold that determines whether the polygon is considered a lake or can be removed. The applied threshold was set to 0.0081 km$^2$ (3 × 3 Landsat pixels). Despite the glacial lake area in Caucasus reaching down to 0.0011 km$^2$ in 1985 and 0.0022 in 2000 [16], i.e., below the size threshold we applied, the minimum polygon size of 3 × 3 pixels can provide more accurate information on whether a lake is classified correctly.

Deep shadows can be a source of noise and misclassification of water on the Landsat images. We used DEMs as an additional information to filter out this type of noise. As a first step, both SRTM and TanDEMx rasters had sinks filled to avoid potential errors. Next, for every lake polygon that fulfilled the size filtering criteria, we computed the statistics based on the DEM properties (elevation). The computed parameters were: standard deviation of elevation and range between minimum and maximum values. The range parameter was used to set a criterion on whether the polygon can be classified as a lake.

To distinguish glacial lakes from other water bodies, a 1-km buffer around the glaciers was used. This makes the dataset of this study comparable with the dataset described in the study by [6]. The glacier outlines used to build the buffer were made by Tielidze and Wheate [47] and derived from the GLIMS/RGI database.

In addition to our different semi-automatic mapping scenarios presented in the following, we produced as reference data set also a manual lake inventory from Landsat 5–7 and 8 data, supported by Google Earth Engine holdings of high-resolution satellite data.

## 4. Method Demonstrations and Results

### 4.1. Detection of Short-Term Changes in Image Time Series

In Section 3.2 we introduce how time series of individual images and water masks derived from them can be used to evaluate changes in water level of a lake by detecting changes in the lake area from the frequency of water existence for individual pixels. Here, we demonstrate the method applying it to the Bashkara lakes (Central Caucasus). These lakes have been closely monitored for potential outburst floods [48]. The lake Bashkara, located to the north of a glacier front, burst out on 1.9.2017 [49]. The area of the lake

decreased from 0.142 km$^2$ on 27.8.2017 to 0.042 km$^2$ on 12.9.2017. In our time series result, the sudden drop in the lake area corresponds well to the date when the GLOF happened (Figure 3). In the Landsat 8 water occurrence/frequency map (lower panel in Figure 3) the distinctly lower water frequency at the southern, eastern and northern margins of Lake Bashkara ((1, right lake) in Figure 3) is due to the partial lake drainage.

The water occurrence/frequency map also demonstrates the filling of a new lake in the area of interest (Figure 3(1.a–1.d)). Lake Lapa ((1, left lake) in Figure 3) developed later than Lake Bashkara, thus showing lower water occurrence.

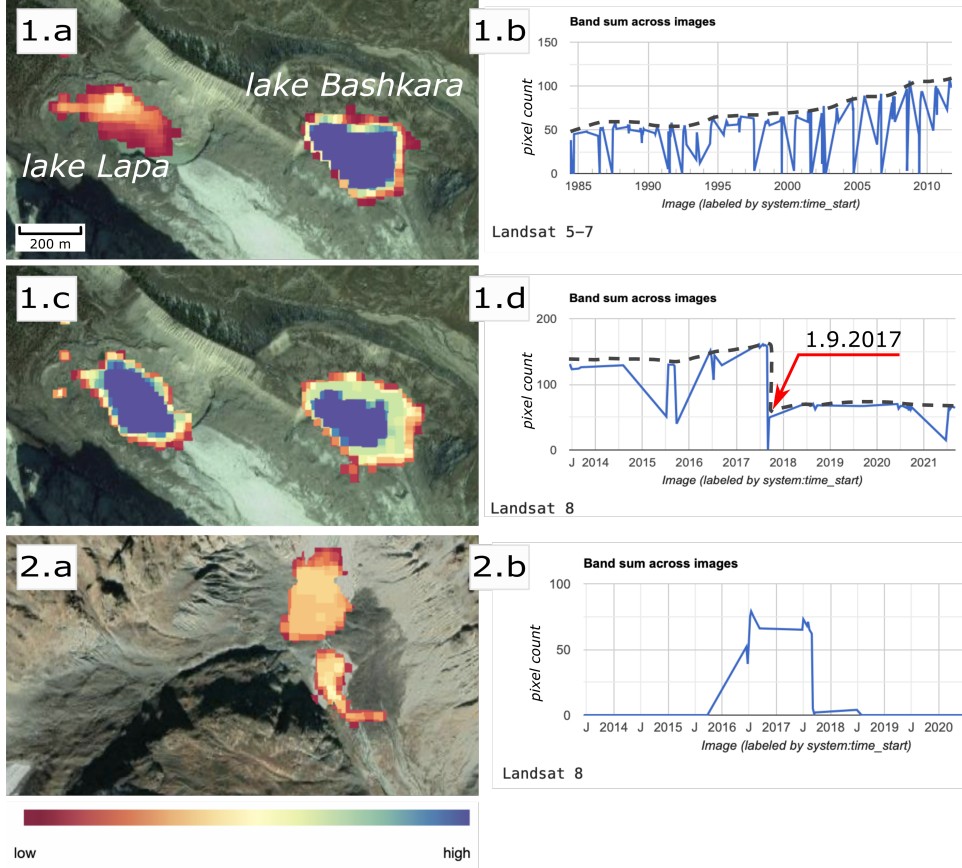

**Figure 3.** Bashkara glacial lakes, Central Caucasus (43° 12′N, 42° 44′E), background image—Google Earth. (**1.a**) Water occurrence/frequency map (how many times pixels are classified as water), source data: Landsat 5 and 7; (**1.b**) time-series of the Bashkara lakes area changes, source data: Landsat 5 and 7; (**1.c**) Water occurrence/frequency map, source data: Landsat 8; (**1.d**) time-series of the Bashkara lakes area changes, the lake Bashkara burst out on 1.9.2017 [49], source data: Landsat 8; The dashed black line corresponds to the actual area changes of the lakes, as an envelope of the local maximum area values; (**2.a**) Water occurrence/frequency map of an unnamed landslide-dammed lake (43°06′N, 42°52′E); (**2.b**) time-series of its area changes. For the location of the example in an overview map see Figure 4.

The approach described allows to manually identify different types of changes in the lake area; whether a lake has burst out, expanded or newly formed. However, there are also various challenges that can be associated with this way of detecting drainages. The pattern of occurrence/frequency depends on the accuracy of classification. Sudden changes in lake area on the time-series curves could also be due to false-positive classification. For example, the sudden drops in the Lake Bashkara time-series curves (Figure 3) correspond to cloudy pixels that cannot be classified as water, regardless the choice of water mapping method, and despite of the actual existence of water below the clouds. In such case, the overall upper bound of the time-series values (upper envelope) is of interest rather than every individual

value. Although, the image collection we used is filtered with the cloud cover percentage indicator before the analysis, some pixels are still occupied by clouds. The size of a Landsat scene is $185 \times 185$ km. For demonstration, if 20% of pixels is covered by clouds, the area that is covered by clouds is still 6660 km$^2$. Moreover, water turbidity and shallow water depth affect spectral properties of the water and can lead to misclassification. Lake turbidity might exactly change in the course of a GLOF, complicating its multispectral detection.

### 4.2. Lake Inventory and Long-Term Changes from Annual Composites

Creating annual composites as described in Section 3.3 and a mosaic of images allows to avoid typical problems that are associated with water mapping, such as seasonal snow cover, clouds, cloud shadows and ice-cover on the lakes. Despite Normalized Difference indices are well recognized as a robust way for mapping the desired surface parameter (vegetation, water, snow etc.) the ratios of the same bands may allow to see more spectral details. The ratios are not normalized, therefore the distribution of pixel values has a higher spread. Ratios pronounce the difference between classes more than normalized differences do. On the negative side, ratios enhance the noise (Figure 1).

We found the min value of $R_{vegetation}$ and max value of $R_{water}$ to be sufficient for glacial lake mapping. Selecting the maximum value of a band representing water ($R_{water}$ bands) represents pixels that can be classified as "water" at least once per time period considered. The minimum value of the "vegetation" normalized difference index or ratio band corresponds to pixels that never contained chlorophyll within a stack of images. For compiling an inventory of glacial lakes in the Greater Caucasus mountain range, the following thresholds have been identified for the above band ratios:

$$Water = R_{water}(max) > 0.2 \,\&\, R_{vegetation}(min) < 0.41 \qquad (8)$$

Taking the minimum of the "vegetation" ratio annual composite and the maximum of the "water" allows one to use two parameters for the annual-scale lake change detection. This avoids the problem caused by lakes that are located at the edge of shadows, where the changing shadow conditions on different individual scenes of the stack would else lead to misclassifications.

Using the annual stack also allows to get rid of artifacts that appear on the water classification of the individual scenes. The thresholds for pixels to be classified as water turns out to be different in different regions. Here, we compared our experiments for the Caucasus with with experiments we conducted for the Everest region, Himalayas (Table 1). The threshold values seem to depend on the water properties, lake depth, turbidity of the water column and the suspended sediment characteristics (likely such as grain size, color and chemical composition), and atmospheric and illumination conditions as far as unaccounted for in the atmospheric correction of the data.

**Table 1.** Example of thresholds, that were applied for annual resolution glacial lake mapping.

|  | Greater Caucasus | Himalaya (Everest Area) |
| --- | --- | --- |
| $R_{water}(max)$ | >0.21 | >0.21 |
| $R_{vegetation}(min)$ | <0.41 | <0.51 |

To increase the accuracy and avoid misclassifications in the shadow areas for the annual inventory of the glacial lakes in the Central Caucasus we also employed the elevation range within lake polygon. The elevation range value within the classified lake polygon depends on the DEM used. For the SRTM elevation model the chosen threshold was 40 m, and 10 m for the TanDEM-X DEM. The threshold values were determined by manually cross-checking the results of the proposed classification method, which is based on band ratios.

For demonstration of method Section 3.3, an inventory of glacial lakes in the Greater Caucasus was created. The ground penetrating radar surveys on several glaciers in the

region done by [50] indicate, that ongoing glacier retreat in the area will cause more lake formation, and pose hazards to the downstream areas. To keep our inventory consistent with previous studies, we focused on lakes that fall within a Landsat frame that covers the Central Caucasus (path: 171, row: 30) (Figure 4).

According to our classification results, there is no trend in the number of lakes detected. Despite there is no strongly pronounced trend in total lake area either, the smallest area of the glacial lakes was observed during the epoch of the data before 1989 and the largest extent was observed in the epoch between 2015 and 2019. However both the number of lakes and the total area have been consistently increasing since the beginning of the 21st century.

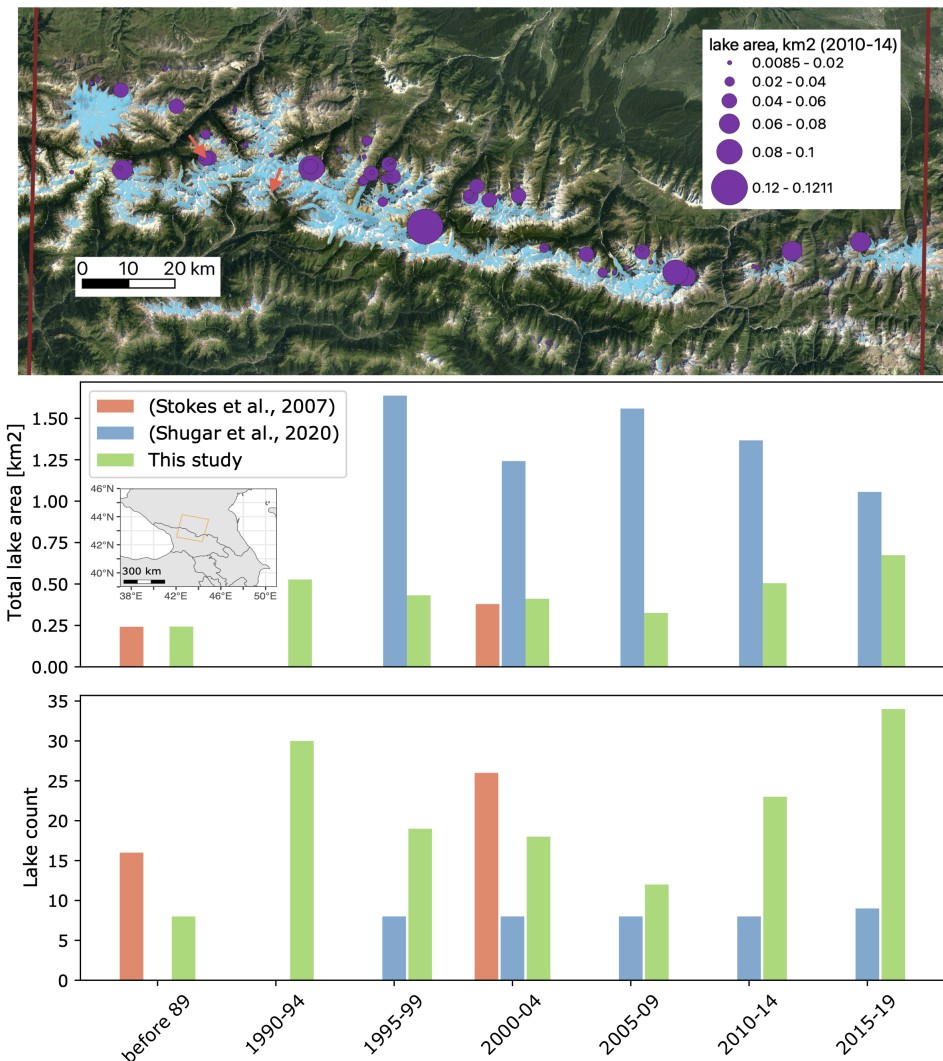

**Figure 4.** Spatial and temporal distribution of the glacial lakes. The results are shown for the footprint of the Landsat frame (path: 171, row: 30). Upper panel: distribution of glacial lakes of the different sizes (2010–2014 epoch). The red line shows the Landsat frame footprint, the blue polygons correspond to the glacier outlines [51] (Background image: Google Earth). Lower panel: comparison of the classification results and existing glacial lake inventories by [6,16]. Red arrows indicate the location of the examples, shown on the Figure 3. The western red arrow indicates the location of Bashkara lake, the eastern one the unnamed landslide lake.

## 5. Discussion

### 5.1. Optical Images vs. Other Input Data Sources

Multispectral optical remote sensing data are the most common type of data used for glacial lake mapping. Several global and regional scale inventories of glacial lakes (including lake drainages and other lake changes) have been produced based on such data (e.g., [6,16,17]). The existing methods usually rely on the combination of thresholds on specific spectral bands and indices. Optical images have several advantages for the purpose of mapping of long-term changes of the high mountain environments. The advantages include the comparably long and consistent record provided by Landsat data. However, the results of water mapping, when done based on individual optical scenes, are also largely influenced by the general issues that affect analyses based on optical images, such as shadows, clouds and cloud shadows. Using SAR data (e.g., [22]) can help to solve some of the problems (such as cloud cover) and increase the temporal resolution due to revisit frequency, however at the expense of shorter time-series available in the archives, and large image parts over mountain areas being affected by geometric effects such as radar shadow, foreshortening and layover. Radar backscatter over lakes varies significantly with variations in waves and thus lake surface roughness.

Although terrain shadows, clouds, cloud shadows and poor data quality might lead to misclassification of the water pixels, the analysis of the time series of lake-area change and the water-occurrence pattern can help to systematically detect glacial lake outburst floods or other flood events in an area and period of interest.

### 5.2. Individual Images vs. Stack of Images

Methods that are based on the analysis of selected individual images are heavily dependent on the availability of good images for the time period of interest. The selection of suitable scenes is very time consuming. Computational platforms such as Google Earth Engine [28] are very well suited for the large-scale regional or global analysis and mapping. The proposed method involves the use of all images that are available for the area of interest and stacks them into annual composites, a procedure that becomes only computationally feasible using massive cloud computing [33]. The use of a large number of scenes allows for consistent datasets to be built. Mosaicking or stacking the images, however, results in "averaged" information over the considered time-period, which reduces the temporal resolution of the results [27]. Data availability also impacts the quality of the mapping. In some areas, very few images are available per year. Therefore some pixels can still be covered by clouds, even after merging them into a composite. The same applies for the noise that comes from data-acquisition errors that affect some scenes. If there are not enough images available, the noise cannot be masked out.

### 5.3. Lake Drainage Detection

The described approach for lake drainage (or damming) detection allows to manually detect different types of changes in the lake area, whether the lake has burst out, expanded, or newly appeared. Despite lake drainages can be initially identified in the time series of lake areas, the proposed method may not be sufficient by itself and does not help to completely overcome the typical problems of optical remote sensing. Various challenges remain. The pattern of occurrence/frequency as well as the lake area curve depend on the accuracy of classification. Despite the lake level drop is reflected in the area changes, the drop in the area curve can correspond to misclassified pixels. Furthermore, some parts of the lake of interest can be obscured by clouds, therefore not being classified as water regardless of the mapping method and resulting in a smaller area of the lake, as shown in the example time-series chart. Moreover, the water turbidity and depth affect spectral properties of the water. Therefore very turbid lakes after a GLOF (such as the example of Bashkara lake on Figure 3d), can be misclassified. The sudden drops in the area time-series, however, frequently only correspond to one or two images, for example with cloud cover or sensor noise, after which the area of the lake "recovers". In the case of an outburst, the

area remains smaller than prior to outburst and the lake does typically not recover on the span of a few weeks, corresponding to the Landsat revisit period. Therefore the proposed method requires manual cross-checking of the result to identify the date of a drainage or damming more precisely and reliably. In our example, the precision with which the timing of an outburst can be detected is set by the revisit time of Landsat. In cases where the detailed outburst time is of interest (which is not necessarily the case for a number of applications) our approach could also be applied to Sentinel-2 data, or other data can visually complement the process, such as daily-resolution PlanetScope data.

Overall, though, the demonstrated application allows clearly to initially identify lake drainages and evaluate lake changes.

### 5.4. Optical Data Alone vs. Combined with DEMs

Most existing inventories of glacial lakes rely on a slope threshold during the classification, derived from digital elevation models in addition to image-derived information. For example, the slope threshold used by [6] equals typically at 40°, or 10°—used by [17]. However, if a elevation model is used, the results depend heavily on the DEM quality.

We compare the results of glacial lake mapping in Central Caucasus (Landsat path: 171, row: 30) that are obtained only using the data cube of annual images (minimum value of "vegatation ratio", band ratio between NIR and Red bands of Landsat image and maximum value of "water ratio"—band ratio between Green and NIR bands—throughout the considered melt season) with results obtained from addition of elevation information (range of the elevation within the classified lake polygon), as derived from both SRTM and TanDEM-X elevation models.

We found that using the elevation information allows to remove a lot of small polygons that are falsely classified as lakes (Figure 5). These polygons pass the size criterion, but correspond to shadowed areas on the slopes instead of actual glacial lakes. These problems could also be reduced by making the thresholds on the spectral information more precise. However, that might on the other end result in a lot of false-negative classifications when applying the same threshold to the stack of images, rather than using a specific threshold for each individual image.

We show that, despite the results possibly being highly dependent on the quality of the DEM, two different DEMs show similar performance when compared with manual mapping from [16] and our own manual cross-checking of the results. The permitted elevation ranges within lake polygons (40 m for the SRTM and 10 m for the TanDEM-X elevation models) that we found to be suitable for the mapping (and filtering of the noise from multispectral based classification) appear reasonable in terms of accuracy of the two DEMs reported [29,31,52–55].

### 5.5. Method Performance: Comparison with Other Studies

We also compare our results with the lake information from existing studies that mapped glacial lakes in our area of interest, such as the regional Central Caucasus study by [16] and global study by [6]. The number of glacial lakes detected using the here-proposed method is higher than proposed by [6] during all 5-year periods considered. Glacial lakes in the Caucasus are smaller than in many other glaciated regions of the world, which makes our study region particularly challenging. According to manual mapping by [16], the lake size in Central Caucasus varied between 0.0022 km$^2$ and 0.0643 km$^2$ in 2000. Therefore, applying the size threshold of 0.05 km$^2$, as is used globally [6] or a size threshold of 0.1 km$^2$ as applied to ice marginal lakes in Greenland by [20], is not suitable for this region. This is also visible in the cumulative lake area plot (Figure 5)—most of the lakes in the area are not captured with the larger size threshold. The locations and total area of the lakes provided by manual mapping by [16] match the locations and area by the manual check of the results of our study (Figure 5).

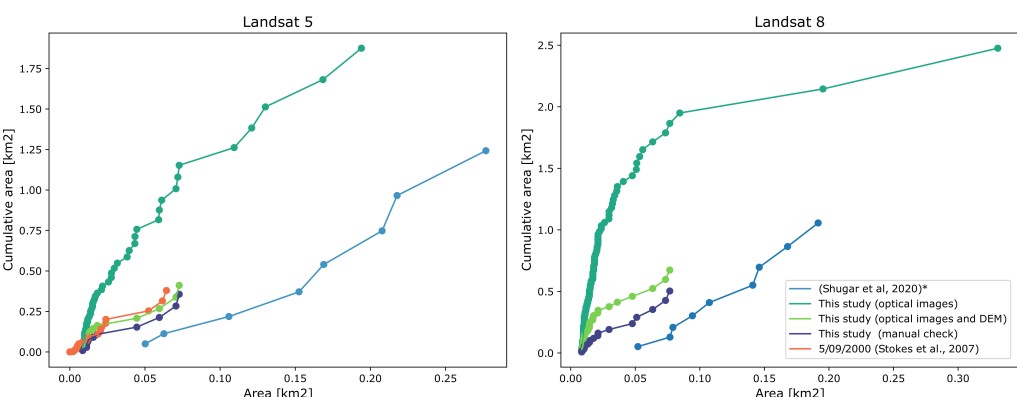

**Figure 5.** The cumulative area of the lakes of different sizes, detected with different methods, suggested in this study, compared with existing glacial lake inventories in Central Caucasus. * Note, the Shugar et al. (2020) study contains substantial misclassifications in the Caucasus.

The large differences in lake numbers and total area between the results of our study and [16] on the one hand, and of [6] on the other hand are mainly due to large misclassifications in the latter study where large areas, for instance on glaciers, are wrongly classified as glaciers.

Using a smaller size threshold to identify lake polygons has, however, some drawbacks. The smaller size increases the amount of noise in the classification (Figure 5). Besides using the terrain information as a way to successfully reduce noise, other parameters of the polygons, such as shape, eccentricity, perimeter and other morphological parameters could be used. However, this should be applied carefully, since glacial lakes can have very different shapes. Pixels that are falsely classified as water could also be filtered with some morphological operations, such as erosion, dilation, opening, closing or median filtering using kernel. Some of these filters reduce the spatial resolution of the data.

## 6. Conclusions

In this study we confirm the potential to map glacial lakes using large stacks of optical data within cloud processing platforms such as Google Earth Engine. We develop several methodological and application scenarios, assess their performance, and discuss potential modifications to our workflows and remaining issues.

While our methodology is able to detect lake drainages with sub-seasonal resolution, does it remain a challenge to identify outburst timing, in particular under poor data availability and cloud cover. Our approach can be applied in addition to other sensors, notably Sentinel-2, or its results be combined with other data types such as SAR data or high-repeat PlanetScope data in order to more accurately pinpoint outburst time - should this be the goal of a study.

Using the minimum and maximum reflection values of a stacked time-series of images allows to eliminate most problems that are associated with the use of optical satellite images for glacial lake mapping, such as ice cover on the lakes, seasonal snow cover, calved icebergs, clouds etc.. However, as a drawback the temporal averaging associated with this min/max stack procedure reduces the temporal resolution of the analysis.

Using band ratios instead of normalized difference indices seems to be the more robust method for glacial lake mapping on an annual time scale because the sensitivity between values for different classes is higher. On the annual scale, the minimum value of the ratio between $Xnir$ and $Xred$ (bands used for the Normalized Difference Vegetation Index (NDVI)) and maximum value of $Xgreen$ and $Xnir$ (bands used for the Normalized Difference Water Index (NDWI)) over the year turns out to be sufficient for glacial lake mapping. The use of band ratios helps to overcome problems from shadows, clouds, water turbidity and suspended sediment concentration. Adding digital elevation model terrain information turns out to be very efficient to remove noise that remains from the multispectral classification and helps thus much to achieve realistic regional scale mapping results.

Inter-comparing three of our own workflows for (pluri-)annual glacier lake mapping (optical images alone, optical images plus DEM, manual), and comparing our results to results from a global-scale automatic mapping study and a regional manual one, shows a large spread of outcomes, both in terms of lake areas and number. This finding suggests that mapping workflows should be carefully parameterized and checked for each mountain range individually in order to account for regional peculiarities such as in input data, spectral properties of glacial lakes and surrounding, topography, lake size and land cover.

**Author Contributions:** Both authors conceptualized and wrote the paper. V.B. performed most analyses. Conceptualization, V.B. and A.K.; Formal analysis, V.B.; Funding acquisition, A.K.; Methodology, V.B. and A.K.; Supervision, A.K.; Visualization, V.B.; Writing—original draft, V.B.; Writing—review & editing, A.K. All authors have read and agreed to the published version of the manuscript.

**Funding:** This research was supported by the ESA project Glaciers CCI (grant no. 4000127593/19/I-NB).

**Conflicts of Interest:** The authors declare no conflict of interest.

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
