# Peer review of "Mapping Area Changes of Glacial Lakes Using Stacks of Optical Satellite Images"

_remotesensing, doi:10.3390/rs14235973_

Round 1

Reviewer 1 Report

Review comments

Review of “MAPPING GLACIAL LAKES USING STACKS OF OPTICAL SATELLITE IMAGES

1.      The authors use simple indices and comparatively low-resolution satellite data to map glacier lakes. The revisit time of Landsat and usually persistent clouds cover in summer season (most important for the glacier lake outbursts) particularly in mountain regions (where GLOFs are the major issue), the replicability of the study is too low. The authors should explore more appropriate satellite data (Sentinel-1 and Sentinel-2) to overcome the mentioned data and make the methodology useful for practical applications.

2.      The authors also agree that the GLOF require continuous monitoring with good temporal resolution as stated in “GLOF monitoring requires good temporal resolution of the data that is used for detection. Analysis of a time series of individual images allows to capture the intra-seasonal changes of the lake water level and area, and the sudden lake area change associated with a GLOF (Fig. 2). The temporal resolution of the Landsat image acquisition is 16 days, therefore, under cloud-free conditions, it allows to identify the date of the event within this timeframe.” Unfortunately, the criteria are not fulfilled by Landsat unless you are too lucky to get cloud-free images. This is again extremely difficult in mountainous regions where sometimes images remain under cloud cover for up to three months or more. In this case, the selection of satellite is totally unconvincing.

·        Line 18-25: too basic for scientific research, I would suggest removing it.

·        Line 35-38: “Furthermore, among other hazards, GLOFs result in the release of large amounts of sediments that travel downstream and can damage, for instance, hydropower and aquaculture infrastructure and agricultural land 38 (e.g. [5], [6]).” It is worth referring to other examples in the most GLOF vulnerable regions e.g. Karakoram/Himalaya.

Author Response

We would like to thank the referees for their comments and suggestions that certainly helped us to clarify our study.

Please find the answer to the comments attached. 

Reviewer 2 Report

Please find my comments in the attached pdf file.

Author Response

(The authors gave the same response as above.)

Reviewer 3 Report

Reviews of the manuscript titled “MAPPING GLACIAL LAKES USING STACKS OF OPTICAL

SATELLITE IMAGES” by Bazilova and Kääb.

Overview:

Mapping glacial lakes is challenging due to their small size and ice cover. The authors mainly used optical satellite dataset on Google Earth Engine to derive the time series image to map glacial lakes and propose a method framework can be applied to detect GLOFs and to produce glacial lake inventory. Overall, this paper is generally well-structured, and the results are clearly presented. However, more detailed descriptions of results are suggested to be strengthened. I just listed some main concerns and suggestions as follows for improving the quality of this paper. 

Major Comments:

1) The study lacks an evaluation of the proposed methodological framework. The comparison between Shugar et al., (2020)/Stokes et al., (2007) shows a large disagreement in Fig. 4, but no explanation is given. Why the authors' method is better than those of previous studies?

2) Figure 3. 1b shows several sudden declines in the lake area, are they all related to GLOF events? Only GLOF events on 1.9.2017 are confirmed?

3) Some super parameters are chosen without sufficient explanation or reasonable evidence. Such as Rwater and Rvegetation.

4) Figure 5 shows the performance from high to low: the methods (optical images and DEM), the methods of Stokes et al., (2007), and the methods of optical images. Did Stokes et al., (2007) also use extra data? Are they manually checked? If not, why their result is better than this study's?

5) The language and writing are nice at the beginning but become worse toward the end of the manuscript.

Line 45 what is the posterior trend? This term is not very common.

Author Response

(The authors gave the same response as above.)

Round 2

Reviewer 1 Report

As mentioned in my previous review, Landsat data quality is insufficient for small glacier lakes due to lower spatial resolution, revisit time, and clouds. I reiterate to use of Sentinel-2 data in this study. Without the suggested revision, I do not recommend it for publication due to its limited scope.
